# Squat Motion of a Humanoid Robot Using Three-Particle Model Predictive Control and Whole-Body Control

**DOI:** 10.3390/s25020435

**Published:** 2025-01-13

**Authors:** Hongxiang Chen, Xiuli Zhang, Mingguo Zhao

**Affiliations:** 1School of Mechanical Electrical and Control Engineering, Beijing Jiaotong University, Beijing 100044, China; 24110374@bjtu.edu.cn (H.C.); zhangxl@bjtu.edu.cn (X.Z.); 2Department of Automation, Tsinghua University, Beijing 100084, China

**Keywords:** humanoid robot, whole body control (WBC), model predictive control (MPC), squat motion

## Abstract

Squatting is a fundamental and crucial movement, often employed as a basic test during robot commissioning, and it plays a significant role in some service industries and in cases when robots perform high-dynamic movements like jumping. Therefore, achieving continuous and precise squatting actions is of great importance for the future development of humanoid robots. In this paper, we apply three-particle model predictive control (TP-MPC) combined with weight-based whole-body control (WBC) to a humanoid robot. In this approach, the arms, legs, and torso are simplified into three particles. TP-MPC is utilized to optimize the rough planning’s reference trajectory, while WBC is employed to follow the optimized trajectory. The algorithm is tested through simulations of a humanoid robot performing continuous squatting motions. It demonstrates the ability to achieve more accurate trajectory tracking compared to using WBC alone and also optimizes the issue of excessive knee torque spikes that occur with WBC alone during squatting. Moreover, the algorithm is less computationally intensive, and it is capable of operating at a frequency of 100 Hz.

## 1. Introduction

### 1.1. Background

In recent years, the development of humanoid robots has been accelerating rapidly, and their motion capabilities have improved remarkably. However, due to the complex dynamics and high degrees of freedom inherent in humanoid robots, coupled with external environmental interferences, achieving precise motion control remains a significant challenge. For service robots, like those utilized in warehousing and logistics, the ability to squat is essential and crucial. These robots have to bend down and squat to handle items, using their arms to perform tasks before standing up again. Traditionally, the control algorithms for bipedal robots have relied on optimization-based methods. Some researchers, such as Kajita et al. have employed the linear inverted pendulum (LIP) method to control bipedal robots on rugged terrains [1]. Xiaobin Xiong et al. have combined LIP with a spring-loaded inverted pendulum (SLIP) to achieve 3D walking in underdriven systems on a particular robot [2]. In 2018, Sangbae Kim et al. applied model predictive control algorithms to footed robot control for the first time [3], and they later proposed the classic MPC + WBC control framework [4], which has had a profound impact on the field.

### 1.2. Motivation

The execution of a squatting motion by a humanoid robot is a complex task that demands intricate whole-body coordination, involving not only the control of the torso’s position and orientation but also the precise positioning of the arms and feet. Moreover, it is crucial to account for the foot force magnitude and adhere to various dynamic constraints. This underscores the complexity of the whole-body control challenge at hand. To address this, we turn to optimization-based whole-body control algorithms.

There are two general approaches to implementing whole-body control algorithms. Hierarchical quadratic programming (HQP) offers a structured hierarchy among tasks, ensuring that tasks are executed without interference. However, this method’s requirement to solve the optimization problem iteratively demands significant computational resources. On the other hand, weighted quadratic programming (WQP) simplifies the process by requiring the optimization problem to be solved only once. Thus, this paper adopts a weight-based quadratic programming approach.

Furthermore, the MPC is often used for trajectory optimization in robot control problems. The study introduces a three-particle model predictive control [5] strategy to humanoid robotics, offering a novel simplification that enhances the accuracy of the control strategy. Conventional MPC approaches often simplify quadrupedal and bipedal robots into a single rigid body model, neglecting the presence of arms and legs, which can be an oversimplification. In contrast, the three-particle model acknowledges the presence of both arms and legs, providing a more nuanced representation. This paper leverages the squatting task as a testbed to explore the merits of the three-particle model, aiming to demonstrate its effectiveness in achieving complex motions with greater accuracy and efficiency. By integrating this approach with whole-body control, we aim to contribute to the advancement of humanoid robot control strategies that are both sophisticated and computationally feasible.

### 1.3. Related Works

There have already been some researchers conducting related studies in this direction. For instance, Francesco Nori and Yue Hu et al. achieved the squatting movement of a humanoid robot by employing a scaled-down version of the iCub bipedal robot along with multiple objective functions [6]. David B. Grimes et al. utilized probabilistic inference in a Bayesian network to explore the problem of a humanoid robot learning the whole-body movement of a human. Based on the HOAP-2 humanoid robot, they realized a squatting movement and a single-leg balancing movement [7]. Yuki Asano et al. designed a screw-home mechanism and applied it to the skeletonoid robot Kenshiro to accomplish continuous twisting squatting movements [8]. Sang-Ho Hyon et al. put forward an imitation learning framework and applied the proposed method to a four-link robot model, successfully realizing squatting actions [9]. David Galdeano et al. proposed a task-based zero-moment point (ZMP) planning, achieved whole-body planning on the humanoid robot HOAP-3, and carried out static balance experiments and squatting experiments under external thrust [10]. Sean Mason et al. proposed a linear quadratic regulator (LQR) for humanoid robots, making use of linearised whole-robot dynamic and contact constraints, and performed push recovery and squat experiments on the Sarcos humanoid robot [11]. Shuuji Kajita et al. implemented forward downward squatting and backward retracting squatting on the HRP-2Kai humanoid robot [12]. L. Penco et al. proposed a redirection method in the research of humanoid whole-body motion. It is a control framework based on inverse kinematics (IK) and a quadratic programming (QP) optimization solver combined with a remote-controlled iCub robot, and the squatting experiments were conducted on an iCub robot [13]. In 2011, Andrej Gams et al. modeled a real person, collected the joint trajectories of a human during squatting, and processed them for the implementation of a continuous squatting action on a bipedal robot. They used the ZMP method to ensure the stability of the robot, applying the idea of zero-space projection and making maintaining stability during the squatting process a top priority [14]. In 2022, J. Duran-Hernandez et al. used the second-order Super Twisting Sliding Mode Control (STSMC) method to achieve squatting on a low-cost bipedal robot and compared it with the proportional integral derivative (PID) method, proving that the method could achieve better trajectory tracking [15]. In 2009, Sang-Ho H et al. proposed a virtual implementation framework for a torque-controlled humanoid robot with a biological musculoskeletal system by integrating a task-space controller and a joint stiffness controller through a superposition method that tested the feasibility of the algorithm using squatting motion [16]. In 2017, Sang-Ho Hyon et al. designed a hydraulic humanoid robot TaeMu, which performed a squatting movement with a sinusoidal signal for the torso trajectory [17]. In 2016, Dingsheng Luo et al. proposed a solution to the problem of non-smooth overshooting when transforming the tasks performed by the robot, converting the problem to a machine-learning approach. The DARwIn-OP robot provided by webots was used in the simulation for the implementation of the training results, and finally, physical experiments were conducted on a PKU-HR5.1 humanoid robot to verify the feasibility of the method [18]. In 2013, Vincent Bonnet et al. proposed a method for estimating the state of a humanoid robot in the sagittal plane during squatting when the joints of the lower limbs as well as the torso are in motion, and they verified the accuracy of the estimation method using squatting motion on the HOAP-3 robot [19]. In the field of exoskeleton robotics, there is also relevant research on squatting. In 2021, Shuzhen L et al. achieved squatting on a lower-limb rehabilitation robot using a reinforcement learning method with strong robustness and anti-interference ability [20]. In 2018, A. N. Miyadaira et al. analyzed the motion during squat jumping for 3DOF articulated legs and proposed two methods to estimate the joint torque change during jumping to optimize the stability of the robot during landing [21].

I-Feng Lee et al. successfully implemented squatting movements on a single robotic leg, providing valuable insights for the study of humanoid robot dynamics [22]. Professor Zhang Liang’s team employed imitation learning algorithms to conduct a phase-time squatting test on the NAO robot [23]. Professor Chen Jianxin’s team achieved lower-body control on a humanoid robot based on gesture information, enabling the robot to perform tasks such as standing on one foot and squatting [24]. In 2022, Wenhan Cai integrated real-time kinematic programming (RKP) and WBC methods to simulate squatting motion on a bipedal robot [25]. In 2023, Pengcheng Lin proposed a method for joint trajectory planning using elastic primitives to enhance the energy storage effect during the landing phase of squatting. This involved conducting human motion capture experiments, data processing, and validating the method’s effectiveness through simulation experiments [26].

### 1.4. Contribution

In this paper, we mainly use the three-particle model predictive control algorithm and the whole-body control algorithm to implement the squatting action on a humanoid robot. The three-particle model predictive control algorithm is used for optimizing the rough trajectories, and the whole-body control algorithm achieves the accurate tracking of the trajectory optimized by the model predictive control and handles the corresponding constraints based on the design of the weights. Ultimately, we achieved more accurate trajectory tracking and optimized the phenomenon of large peak knee moments compared to the WBC-only approach. In addition, this model simplification allows the MPC problem to be solved within 5.5 ms per frame, in most cases, which greatly improves the computational efficiency.

The rest of the paper is organized as follows: the second part introduces the three-particle model predictive control and whole body control algorithm, the third part introduces the results and discussion, and the fourth section concludes the paper as well as provides some directions for the future.

## 2. Methods

### 2.1. Three-Particle Model Predictive Control

#### 2.1.1. Modeling

As shown in Figure 1, the humanoid robot used in this paper has a mass of 20.5 kg and a total of 19 degrees of freedom. A single arm has 3 degrees of freedom, a single leg has 6 degrees of freedom, and the chest has 1 degree of freedom, all of which are rotary joints. Considering the characteristics of the squatting motion and the requirement for the real-time operation of the algorithm, we simplify the humanoid robot into three particles. Specifically, the two arms, two legs, and the torso are simplified into three particles in a three-dimensional space. The center of mass (CoM) of the torso is calculated from the central trunk position and the chest above; the CoM of the two legs and the two arms is calculated from the vector sum of the positions of the end joints and the hip (shoulder) joints to find the midpoints. The end position information is measured by the sensors in the simulation, and the hip (shoulder) joint position is obtained from the sensors at the torso by adding the respective biases. For example, the CoM of a single leg is given by(1)$$p_{m,leg}=\frac{1}{2}\left[\begin{array}{c} p_{b}+l_{hip}+p_{foot} \end{array}\right]$$
where $p_{m,leg}$ denotes the CoM position of a single leg, $p_{b}\in R^{3\times 1}$ denotes the trunk position, $l_{hip}\in R^{3\times 1}$ denotes hip offset relative to the trunk position, and $p_{foot}\in R^{3\times 1}$ denotes the foot position. The CoM of the torso is obtained as follows:(2)$$p_{m,torso}=\frac{m_{b1}p_{m,b1}+m_{b2}p_{m,b2}}{m_{b1}+m_{b2}}$$
where $m_{b1}$ denotes the central trunk mass, $m_{b2}$ denotes the chest mass, $p_{m,b1}$ denotes the CoM of thr trunk, and $p_{m,b2}$ denotes the CoM of the chest.

The final overall simplified CoM is obtained as follows:(3)$$\begin{array}{c} p_{CoM}=\frac{m_{leg}p_{m,leg}+m_{arm}p_{m,arm}+m_{b}p_{m,torso}}{m} \end{array}$$
where $p_{CoM}$ denotes the CoM position and $m=m_{b}+m_{leg}+m_{arm}$ is the overall mass. The positions and velocities of the arm ends, leg ends, and torso are selected as the system’s state variables, with their respective accelerations serving as control inputs to establish the discrete state-space equations as follows:(4)$$x_{k+1}=\bar{A}x_{k}+\bar{B}u_{k}$$
where $x_{k}={\left[\begin{array}{cccccc} p_{b}^{T} & p_{lh}^{T} & p_{rh}^{T} & {\dot{p}}_{b}^{T} & {\dot{p}}_{lh}^{T} & {\dot{p}}_{rh}^{T} \end{array}\right]}^{T}$, and it includes the position and velocity of the torso, the end of the left hand, and the end of the right arm. $u_{k}={\left[\begin{array}{ccc} {\ddot{p}}_{b}^{T} & {\ddot{p}}_{lh}^{T} & {\ddot{p}}_{rh}^{T} \end{array}\right]}^{T}$, and it contains the acceleration of the torso, the end of the left hand, and the end of the right arm. $\bar{A}$ is the state matrix and $\bar{B}$ is the control matrix. In addition, the states of the simplified CoM can be deduced by combining Equations (Equation 1)–(Equation 3) as follows:(5)$$p_{CoM,k}=Cx_{k}+D$$(6)$${\dot{p}}_{CoM,k}=Ex_{k}$$
after the N-step prediction, the compact form of the state equation is as follows:(7)$$X=A_{qp}x_{0}+B_{qp}U$$
among them,(8)$$A_{qp}=\left[\begin{array}{c} \bar{A}\\ {\bar{A}}^{2}\\ \vdots \\ {\bar{A}}^{N} \end{array}\right]\in R^{18N\times 18}$$(9)$$B_{qp}=\left[\begin{array}{ccccc} \bar{B} & 0 & \cdots & 0 & \\ \bar{A}\bar{B} & \bar{B} & \cdots & 0\\ \vdots & \vdots & \ddots & \vdots & \\ {\bar{A}}^{N-1}\bar{B} & {\bar{B}}^{N-2}\bar{B} & \cdots & \bar{B} \end{array}\right]\in R^{18N\times 9N}$$
$x_{0}$ denotes the current state of the system that needs to be acquired each time the MPC problem is computed, and U denotes the optimal control sequence that needs to be computed at the current torque. Correspondingly, the states of the simplified CoM are represented as follows:(10)$$P_{CoM}=\bar{C}X+\bar{D}$$(11)$${\dot{P}}_{CoM}=\bar{E}X$$ Correspondingly,(12)$$\begin{array}{c} \bar{C}=\left[\begin{array}{ccc} C & & \\ & \ddots & \\ & & C \end{array}\right]\in R^{3N\times 18N}\\ \bar{D}=\left[\begin{array}{c} D\\ \vdots \\ D \end{array}\right]\in R^{3N\times 1}\\ \bar{E}=\left[\begin{array}{ccc} E & & \\ & \ddots & \\ & & E \end{array}\right]\in R^{3N\times 18N} \end{array}$$
where the matrices C and E and the vector D are used to calculate the state quantities of the simplified center of mass.

#### 2.1.2. Cost Function

The significance of establishing the cost function is to ensure that the trajectory following error during squatting is as small as possible and the amount of control required is not too large, so the following cost function is established:(13)$$\begin{array}{c} J=\min_{U}{∥P_{CoM}-P_{CoM}^{ref}∥}_{\bar{Q}}+{∥{\dot{P}}_{CoM}-{\dot{P}}_{CoM}^{ref}∥}_{\bar{R}}+{∥{\bar{P}}_{arm}-{\bar{P}}_{arm}^{ref}∥}_{\bar{S}}+{∥U∥}_{\bar{T}} \end{array}$$ The position and velocity tracking terms of the simplified CoM position are ${∥P_{CoM}-P_{CoM}^{ref}∥}_{\bar{Q}}$ and ${∥{\dot{P}}_{CoM}-{\dot{P}}_{CoM}^{ref}∥}_{\bar{R}}$, the position and velocity tracking term of the end of the arm is ${∥{\bar{P}}_{arm}-{\bar{P}}_{arm}^{ref}∥}_{\bar{S}}$, and the penalty term of the control input ${∥U∥}_{\bar{T}}$ is included in Equation (Equation 13). In Equation (Equation 13), the weight of the CoM position tracking term $\bar{Q}\in R^{3N\times 3N}$, the weight of CoM velocity tracking term $\bar{R}\in R^{3N\times 3N}$, the weight of end-of-arm position and velocity tracking term $\bar{S}\in R^{12N\times 12N}$, and the weight of control penalty term $\bar{T}\in R^{3N\times 3N}$, respectively.

The corresponding weighting factors are shown in Table 1.

Due to the excessive length of the third term, we split it into separate representations for the left and right arms.

The cost function is further converted to a QP form by combining Equations (Equation 7) and (Equation 13) as follows:(14)$$\min_{U}\frac{1}{2}U^{T}HU+U^{T}g$$(15)$$H=H_{1}+H_{2}+H_{3}+H_{4}$$(16)$$g=g_{1}+g_{2}+g_{3}+g_{4}$$
where H,g denotes the sum of the coefficients of the CoM position and velocity tracking, the arm end position and velocity tracking, and the input penalty term task. For example, $H_{1}$, $g_{1}$, $H_{2}$, $g_{2}$ are the task matrices and vectors obtained when deriving the CoM position and velocity following term into the standard QP form.(17)$$H_{1}=2B_{qp}^{T}\bar{C^{T}}\bar{Q}\bar{C}B_{qp}$$(18)$$g_{1}=2B_{qp}^{T}\bar{C^{T}}\bar{Q}\left(-p_{CoM,pos}^{ref}+\bar{C}A_{qp}x_{0}+\bar{D}\right)$$(19)$$H_{2}=2B_{qp}^{T}\bar{E^{T}}\bar{R}\bar{E}B_{qp}$$(20)$$g_{2}=2B_{qp}^{T}\bar{E^{T}}\bar{R}\left(-p_{CoM,vel}^{ref}+\bar{E}A_{qp}x_{0}\right)$$
where $p_{CoM,pos}^{ref}$, $p_{CoM,vel}^{ref}$ are the reference trajectories of the simplified CoM obtained by rough planning.

This is finally solved using the qpOASES library, and the first frame of data obtained from the optimization solution is used as the reference input for the WBC, The overall control block diagram is shown in Figure 2.

### 2.2. Whole-Body Control

#### 2.2.1. Modeling

The humanoid robot is generally modeled as a full-dynamics model containing a floating base when the whole-body control method is used, in which case, the generalized joints of the robot are expressed as follows:(21)$$q=\left[\begin{array}{c} q_{f}\\ q_{j} \end{array}\right]\in R^{n_{q}}$$
where $q_{f}\in R^{6}$ is used to describe the position and attitude of the robot torso and the attitude is generally expressed in Euler angles or quaternion; $q_{j}\in R^{n_{j}}$ is used to describe the joint angles of all real joints, and(22)$$n_{q}=n_{j}+6$$ The full dynamic equation is expressed as follows:(23)$$M\left(q\right)\ddot{q}+h\left(q,\dot{q}\right)=\left[\begin{array}{c} 0_{6\times 1}\\ \tau_{j} \end{array}\right]+J_{c}^{T}\left(q\right)\omega_{c}$$
where $\dot{q}$ is the generalized joint velocity; $\ddot{q}$ is the generalized joint acceleration; $M\left(q\right)\in R^{n_{j}\times n_{j}}$ is the mass inertia matrix dependent on the q, $h(q,\dot{q})$ and contains the Coriolis, centrifugal, and gravity terms; $\tau_{j}$ is the real joint driving torque; $J_{c}\left(q\right)\in R^{12\times n_{q}}$ is the contact Jacobian matrix; and $\omega_{c}={\left[\begin{array}{cc} \omega_{c1}^{T} & \omega_{c2}^{T} \end{array}\right]}^{T}$ represents the contact wrenches, where $\omega_{c1}^{T}={\left[\begin{array}{cc} f_{1}^{T} & \tau_{1}^{T} \end{array}\right]}^{T}\in R^{6\times 1}$ denotes the end-foot force and torque of the contact foot.

#### 2.2.2. Whole-Body Control Formulation

There are various forms of implementing the full-body control algorithms, and building a quadratic programming problem containing weights is one of them. Its standard form is as follows:(24)$$\begin{array}{c} \underset{\chi }{\arg\min}\sum_{i=1}^{n_{task}}{∥W_{i}\left(A_{i}\chi -b_{i}\right)∥}_{2}^{2}\\ s.t.\;lb_{k}\leq C_{k}\chi \leq ub_{k}(k=1,2,...k_{constraint}) \end{array}$$
where $A_{i}\;B_{i}$ denotes the task matrix and task vector of the corresponding task; $W_{i}$ is the weight of the corresponding task; $\chi ={\left[\begin{array}{cc} {\ddot{q}}^{T} & \omega^{T} \end{array}\right]}^{T}$ is the optimization variable, including the generalized joint acceleration and wrenches; $C_{k}$ is the constraint matrix of the corresponding constraint; and $lb_{k}\;ub_{k}$ is the lower limit and upper limit of the corresponding constraint. Finally, it is converted into an optimization problem to solve the optimization variables, and the joint torque $\tau_{j}$ can be obtained by combining it with the full dynamics Equation (Equation 17). The robot needs to coordinate the whole-body joint movement to achieve the squatting action, so the following subtasks and constraints are set as shown in the Table 2.

The weighting factors for each task are shown in Table 3.

#### 2.2.3. Tasks

##### Torso Trajectory Tracking Task

It is important to achieve the accurate trajectory tracking of the torso during the squat, so the torso task was set up as follows:(25)$$\left[\begin{array}{cc} I_{6\times 6} & 0_{6\times (n_{j}+12)} \end{array}\right]\chi -\left[\begin{array}{c} {\ddot{p}}_{torso}^{des}\\ \ddot{\Theta_{torso}^{des}} \end{array}\right]$$
where χ is the variable to be optimized; $I_{6\times 6}$ is a unit diagonal matrix used to select the optimization variables required for this task; and ${\ddot{p}}_{torso}^{des},{\ddot{\Theta }}_{torso}^{des}$ are the desired torso linear acceleration and angular acceleration, obtained from the PD controller as follows:(26)$$\begin{array}{c} {\ddot{x}}_{torso}^{cmd}={\ddot{x}}_{torso}^{des}+k_{p}\left(x_{torso}^{des}-x_{torso}^{fb}\right)+k_{d}\left({\dot{x}}_{torso}^{des}-{\dot{x}}_{torso}^{fb}\right) \end{array}$$ Here $x_{torso}^{des},\;{\dot{x}}_{torso}^{des},\;{\ddot{x}}_{torso}^{des}$ are obtained by TP-MPC.

##### Arm Trajectory Tracking Task

The arms need to follow the torso during the squat, so we set up the arm task as follows:(27)$$\left[\begin{array}{cc} J_{hand}\left(q^{fb}\right) & 0_{6\times 12} \end{array}\right]\chi -\left[\begin{array}{cc} {\ddot{x}}_{hand}^{des} & -{\dot{J}}_{hand}\left(q^{fb}\right){\dot{q}}^{fb} \end{array}\right]$$
where ${\ddot{x}}_{hand}$ contains $\begin{array}{cc} {\ddot{p}}_{hand}^{des} & {\ddot{\Theta }}_{hand}^{des} \end{array}$, which represent the desired linear and angular accelerations of the arm; $q^{fb}$ and ${\dot{q}}^{fb}$ are the generalized joint position and velocity feedback, respectively; and $J_{hand}\in R^{6\times n_{q}}$ is the arm Jacobian matrix.

##### Foot Trajectory Tracking Task

The feet need to remain stationary relative to the ground during the squat, so we set up the feet task as follows:(28)$$\left[\begin{array}{cc} J_{foot}\left(q^{fb}\right) & 0_{6\times 12} \end{array}\right]\chi -\left[\begin{array}{cc} {\ddot{x}}_{foot}^{des} & -{\dot{J}}_{foot}\left(q^{fb}\right){\dot{q}}^{fb} \end{array}\right]$$
where ${\ddot{x}}_{foot}$ contains $\begin{array}{cc} {\ddot{p}}_{foot}^{des} & {\ddot{\Theta }}_{foot}^{des} \end{array}$, which are the desired linear and angular acceleration of the foot; $q^{fb}$ and ${\dot{q}}^{fb}$ are the generalized joint position and velocity feedback, respectively; and $J_{foot}\in R^{6\times n_{q}}$ is the foot Jacobian matrix.

##### Chest Trajectory Tracking Task

The squatting process does not require the chest joint, so this joint is fixed to the zero position and the chest joint task is set as follows:(29)$$\left[\begin{array}{ccc} 0_{1\times 6} & I & 0_{1\times 30} \end{array}\right]\chi -\left[\begin{array}{c} 0 \end{array}\right]$$The chest joint expectation is set to zero in this task.

##### Wrench Task

To ensure that contact forces are not excessive during the squat, the task is formulated as follows:(30)$$\left[\begin{array}{cc} 0_{12\times (n_{j}+6)} & I_{12\times 12} \end{array}\right]\chi -\left[\begin{array}{c} 0_{12\times 1} \end{array}\right]$$

#### 2.2.4. Constraints

In the process of solving the optimization problem, it is necessary to set the appropriate constraints; otherwise, the results will be solved, which cannot be achieved in reality, so the following constraints are set:

##### Floating Base Dynamics Constraint

In the full dynamics model, we often describe the 6-dimensional motion possessed by the floating base as the result of being driven by a virtual joint containing 6 DoF, so we need to ensure that the actual driving force of these 6 DoF is zero when solving the optimization, and we therefore set this constraint as follows:(31)$$\left[S_{f}M\left(q^{fb}\right)\;-S_{f}J_{foot}^{T}\left(q^{fb}\right)\right]\chi =-S_{f}h\left(q^{fb},{\dot{q}}^{fb}\right)$$
where $S_{f}=\left[\begin{array}{c} I_{6\times 6}\;0_{6\times n_{j}} \end{array}\right]$ is the floating base selection matrix used to select the virtual joints mentioned earlier.

##### Contact Force Constraint

In order to ensure that the robot foot end does not slip, it should be ensured that the foot end force is within the cone of friction, and also that the foot sole force in the vertical direction should not be too large, so the constraint is formulated as follows:(32)$$\begin{array}{c} \left|f_{x}\right|\leq \mu f_{z}\\ \left|f_{y}\right|\leq \mu f_{z}\\ 0\leq f_{z}\leq f_{z\max} \end{array}$$
where μ is the coefficient of friction and $f_{z}$ is the maximum contact force in the z direction. We describe Equation (Equation 32) as a form of Equation (Equation 24) as follows:(33)$${\mathrm{lb}}_{f}\leq C_{f}\chi \leq {\mathrm{ub}}_{f}$$ The corresponding constraint matrix with upper and lower bounds is denoted as follows:(34)$$C_{f}=\left[\begin{array}{ccccc} 0_{5\times n_{q}} & C_{1,f_{5\times 3}} & 0_{5\times 3} & 0_{5\times 3} & \\ 0_{5\times n_{q}} & 0_{5\times 3} & C_{2,f_{5\times 3}} & 0_{5\times 3} \end{array}\right]$$(35)$$\begin{array}{c} lb_{f}=0_{10\times 1}\\ ub_{f}=\infty_{10\times 1} \end{array}$$(36)$$C_{1,f}=C_{2,f}=\left[\begin{array}{ccc} 1 & 0 & \mu \\ -1 & 0 & \mu \\ 0 & 1 & \mu \\ 0 & -1 & \mu \\ 0 & 0 & 1 \end{array}\right]$$

##### Joint Torque Constraint

In order to ensure the availability of the joint moments, it is necessary to limit the joint moments obtained from the optimization solution to a certain range. We morph Equation (Equation 23) into the form of Equation (Equation 24), from which we can obtain the correlation matrix for this constraint as follows:(37)$$\tau_{j}-S_{j}h\left(q^{fb},{\dot{q}}^{fb}\right)=S_{j}\left[M\left(q^{fb}\right)\;-{J_{foot}}^{T}\left(q^{fb}\right)\right]\chi$$(38)$${\mathrm{lb}}_{\tau }\leq C_{\tau }\leq {\mathrm{ub}}_{\tau }$$(39)$$C_{\tau }=S_{j}\left[M\left(q^{fb}\right)\;-{J_{foot}}^{T}\left(q^{fb}\right)\right]$$(40)$$lb_{\tau }=\tau_{\min}-S_{j}h\left(q^{fb},{\dot{q}}^{fb}\right)$$(41)$$ub_{\tau }=\tau_{\max}-S_{j}h\left(q^{fb},{\dot{q}}^{fb}\right)$$
where $S_{j}=\left[\begin{array}{c} 0_{n_{j}\times 6}\;I_{n_{j}\times n_{j}} \end{array}\right]$ is the joint torque selection matrix used to select the drive joints in the optimized variables. $\tau_{\max}$ and $\tau_{\min}$ are the actual maximum and minimum joint torque values.

##### Joint Velocity Constraint

In order to ensure the effectiveness of the joint rotational speed, it is necessary to limit the joint rotational speed to a certain range. We morph Equation (Equation 23) into the form of Equation (Equation 24), from which we can obtain the correlation matrix for this constraint as follows:(42)$$\dot{q}\left(t+\Delta t\right)=\dot{q}\left(t\right)+\Delta t\ddot{q}\left(t\right)$$(43)$${\mathrm{lb}}_{{\dot{q}}_{j}}\leq C_{{\dot{q}}_{j}}\leq {\mathrm{ub}}_{{\dot{q}}_{j}}$$(44)$$C_{{\dot{q}}_{j}}=\left[0_{n_{j}\times 6}\;\Delta tI_{n_{j}\times n_{j}}\;0_{n_{j}\times 12}\right]$$(45)$$lb_{{\dot{q}}_{j}}={\dot{q}}_{j_{\min}}-{\dot{q}}_{j}^{fb}$$(46)$$ub_{{\dot{q}}_{j}}={\dot{q}}_{j_{\max}}-{\dot{q}}_{j}^{fb}$$

##### Joint Power Constraint

In order to ensure the long and stable operation of the joint, it is necessary to keep the operating power within the normal range. We morph Equation (Equation 23) into the form of Equation (Equation 24), from which we can obtain the correlation matrix for this constraint as follows:(47)$$\tau_{j}-S_{j}h\left(q^{fb},{\dot{q}}^{fb}\right)=S_{j}\left[M\left(q^{fb}\right)\;-{J_{foot}}^{T}\left(q^{fb}\right)\right]\chi$$(48)$${\mathrm{lb}}_{p}\leq C_{p}\leq {\mathrm{ub}}_{p}$$(49)$$C_{p}=diag\left({\dot{q}}^{fb}\right)S_{j}\left[M\left(q^{fb}\right)\;-{J_{foot}}^{T}\left(q^{fb}\right)\right]\chi$$(50)$$\begin{array}{c} lb_{P}=-\infty \end{array}$$(51)$$ub_{P}=P_{j\max}-diag\left({\dot{q}}^{fb}\right)S_{j}h\left(q^{fb},{\dot{q}}^{fb}\right)$$
where $P_{j\max}$ is the joint maximum power.

### 2.3. Experimental Design for Simulation

The humanoid robot used is shown in Figure 3. The simulation experiments were conducted using the open source simulation software Webots, it is an open source and multi-platform desktop application for simulating robots. It provides a complete development environment to model, program, and simulate robots. The QP problem was solved using the qpOASES library [27], it is an open source C++ library specializing in solving small- and medium-sized convex quadratic programming problems. Favored for its efficient performance and guaranteed numerical stability, qpOASES is especially suited for areas such as robot motion planning, machine learning, and control. The dynamics and kinematics were resolved using the rigid body dynamics library (RBDL), which is a high-performance C++ library for rigid body dynamics modeling. It also includes some important rigid body dynamics algorithms, such as ABA, RNEA, CRBA, etc, which enable efficient operations on joint space inertia matrices [28].

For the determination of the trajectory of the humanoid robot when squatting, we conducted motion capture experiments. We used the Nokov optical 3D motion capture system, which can capture the reflective markers placed on the target to accurately capture the 3-dimensional spatial position. Through advanced algorithms used for processing and computing, the system can get the 3-dimensional spatial coordinates of the reflective markers in different time units of measurement (X, Y, Z); it can also be set up on the target object’s rigid body. The system can also be set up on the rigid body of the target object, and through the professional analysis software it can further process and calculate the data, obtaining the three-dimensional data of the target object such as the precise position and attitude. We obtained the trajectory of the X, Y, and Z directions of the real person when squatting through the motion capture equipment, as shown in Figure 4. Through the motion capture data, it can be observed that the torso not only reciprocates in the z-direction but also has a brief reciprocal displacement in the x-direction, and the trajectory of the torso is similar to the triangular wave in the process of squatting. Moreover, the trajectory is similar to the square wave if there is a short pause in the squatting of the two limiting positions, so we used the periodic triangular and square wave trajectories for the squatting test. In the simulation inspired by the squatting trajectories obtained from human motion capture, we first describe the trajectories in our experiments as square and triangular waves. Then, the trajectory is optimized using the three-particle model predictive control, and the optimized trajectory is given for the whole-body control to follow. In some related studies in the past, researchers primarily used the WBC method alone to realize some functions, such as push-recovery [29]. Therefore, we designed a comparative experiment to track the square and triangular wave trajectories using only the whole-body control method. The trajectory period in all experiments was set to 2 s and the squat height was 0.2 m.

## 3. Results and Discussion

### 3.1. Square Wave

Two sets of tests are set up in the simulation, and the reference trajectories are both square wave trajectories as well, as shown in Figure 5. Here, the blue dashed line is the reference trajectory, the red solid line is the trajectory of the squatting motion realized by the WBC-WQP control, and the yellow solid line is the trajectory curve obtained after adding the TP-MPC.

As can be observed in Figure 5, the tracking effect of the WBC-only model has an obvious lag effect and a large tracking error, which makes the tracking effect poor overall. However, the tracking effect finally achieved by adding the TP-MPC is better than that of the WBC-only algorithm for the WBC algorithm, which only considers the current moment. Alternatively, the MPC observes the reference state in the future segment of the time domain and thus optimizes the coarsely planned trajectory into a feasible trajectory that allows WBC to follow the trajectory completely.

Furthermore, during the squatting process, the knee joint plays a crucial role; as shown in Figure 6, it can be observed that a large torque spike occurs when only the WBC algorithm is used, which is extremely bad for joint motors. However, the spikes in the knee joint moments are obviously significantly weakened when the TP-MPC is added. The reason may be that when the reference trajectory is a square wave, there is a step between the position and the velocity, which leads to the existence of obvious spikes in the calculated moments, but when TP-MPC is added, the rough trajectory is optimised to a smoother trajectory, and the final optimised knee moments for tracking the smoother trajectory no longer have large spikes.

The real-time nature of the algorithms also needs to be guaranteed. As shown in Figure 7, the algorithm is able to compute within 5.5 ms in most cases and within 10 ms overall, which has a high computational efficiency. This high computational efficiency may be due to the simplification of the three-particle model used, as mentioned before.

Finally, we calculated the trajectory tracking error in each direction with Equation (Equation 52); we calculated the mean square error throughout the simulation. The trajectory tracking errors of the two algorithms when the reference trajectory is a square wave are shown in Table 4.
(52)$$\mathrm{MSE}=\frac{1}{N}\sum_{i=1}^{N}{\left(y_{i}-{\overset{⌢}{\;y}}_{i}\right)}^{2}$$
where N is the total amount of data, $y_{i}$ is the reference value, and ${\overset{⌢}{\;y}}_{i}$ is the feedback value. From Table 4, it can be observed that the overall tracking error is small, compared to the control framework with TP-MPC + WBC, which is more capable of achieving accurate trajectory tracking.

### 3.2. Triangle Wave

When the reference trajectory is a triangle wave, the tracking effect is as shown in Figure 8; the period of the reference trajectory is 2 s and the squatting height is 0.2 m, which is consistent with the square wave, and the same phenomenon can be observed from the local zoomed-in figure. Only the WBC realizes the squatting with an obvious lag phenomenon, while being far away from the reference trajectory, and the obvious tracking error contrast can be observed, especially in the y-direction.

As shown in Figure 8, when TP-MPC is added, the trajectory tracking accuracies in the x, y, and z directions are all improved due to the fact that TP-MPC is able to take into account the reference trajectories in the future time domain in advance.

The knee torque when the reference trajectory is a triangular wave is shown in Figure 9, and it can be observed that the phenomenon of large torque peaks has been somewhat reduced, which is also a result of the triangular wave being optimized to be relatively smooth.

When the reference trajectory is a triangular wave, the computational time of the MPC also mostly stays within 5.5 ms. It also shows high computational efficiency, as shown in Figure 10.

Finally, the trajectory tracking errors of the two algorithms when the reference trajectory is a triangle wave are shown in Table 5. From the data, it can be seen that the squatting motion implemented by the TP-MPC + WBC framework has a smaller tracking error compared to the WBC alone.

## 4. Conclusions

In this paper, the TP-MPC is applied to a humanoid robot and combined with the WBC algorithm to achieve continuous squatting movements. TP-MPC optimizes the rough planning trajectory into a smoother reference path, while WBC strictly follows the optimized reference trajectory. Additionally, comparison experiments were conducted between the combination of TP-MPC and WBC and WBC alone to achieve continuous squatting, using square wave and triangle wave reference trajectories. Both sets of experiments confirmed that the trajectory tracking effect achieved by combining this method (TP-MPC + WBC) was superior to that of WBC alone. The phenomenon of larger spikes in knee torque during squatting, which occurred when only using WBC, was also optimized when the TP-MPC algorithm was included. Furthermore, the algorithm is computationally efficient, with the overall calculation completed within 10 ms. However, TP-MPC has some limitations. The algorithm simplifies the humanoid robot into a particle model and only considers kinematics, which means it cannot handle the robot’s pose information and has poor resistance to external dynamic disturbances. In the future, it may be necessary to continue improving the algorithm and attempt to incorporate dynamics to realize a more optimal control strategy.

Furthermore, reinforcement learning (RL) is gaining increasing popularity in the field of robot control. Consequently, employing RL methods could be considered for achieving squatting movements in humanoid robots and to compare the experimental outcomes with those of traditional control methods, followed by an analysis.

## Figures and Tables

**Figure 1 sensors-25-00435-f001:**
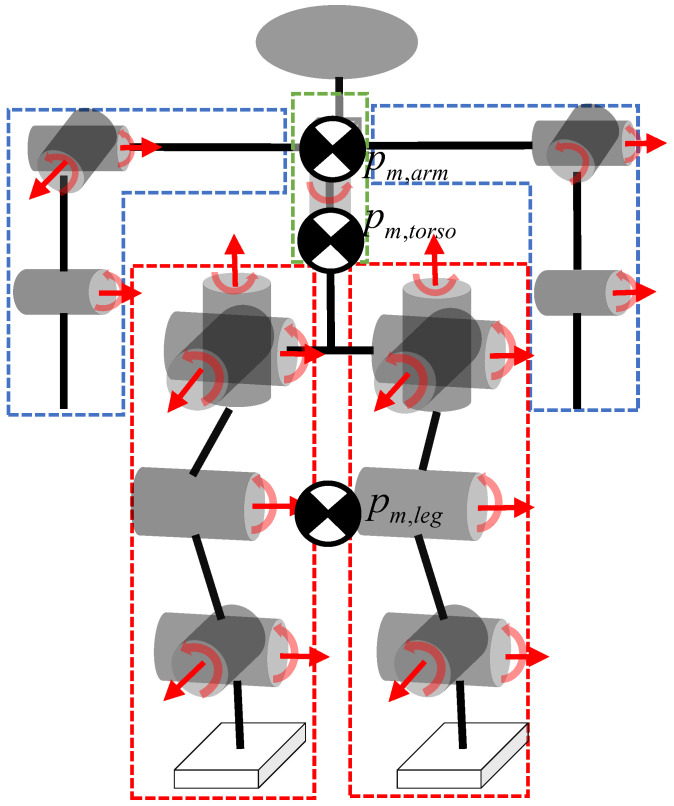
The three-particle model of humanoid robots.

**Figure 2 sensors-25-00435-f002:**
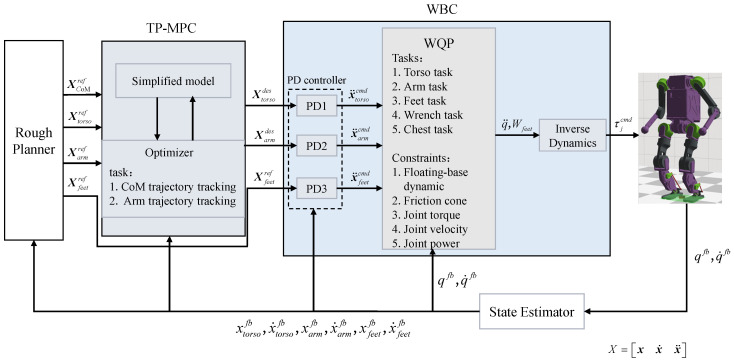
The control framework for the squatting motion of a humanoid robot based on TP-MPC + WBC.

**Figure 3 sensors-25-00435-f003:**
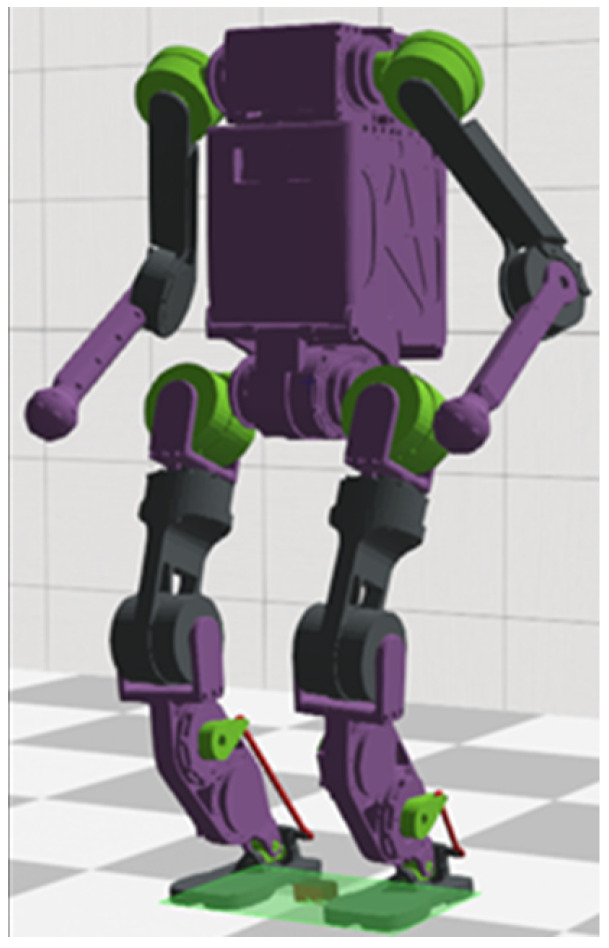
Modeling of the humanoid robot.

**Figure 4 sensors-25-00435-f004:**
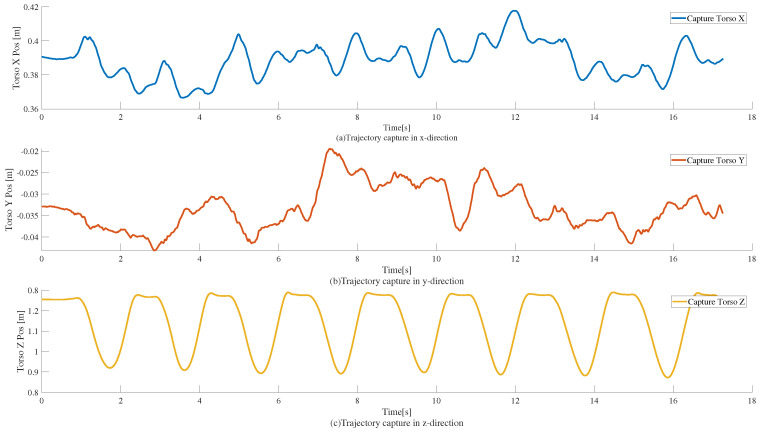
Trajectories of motion capture during human squatting exercise. Here, (**a**–**c**) represent the trajectories in the x, y, and z directions, respectively.

**Figure 5 sensors-25-00435-f005:**
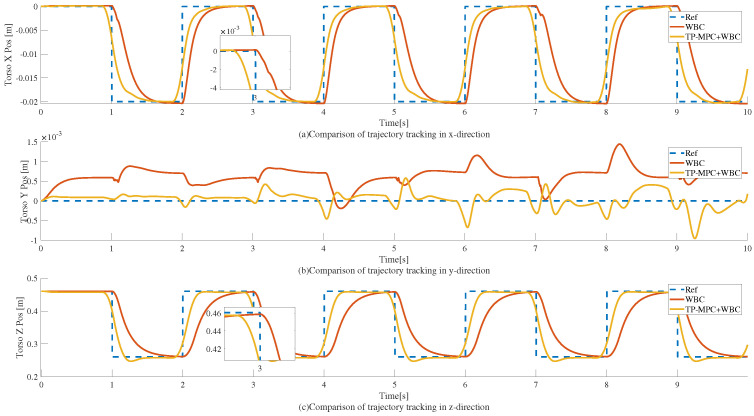
Trajectory tracking of a humanoid robot squatting motion when the reference trajectory is a square wave. Here, (**a**–**c**) represent the trajectories in the x, y, and z directions, respectively.

**Figure 6 sensors-25-00435-f006:**
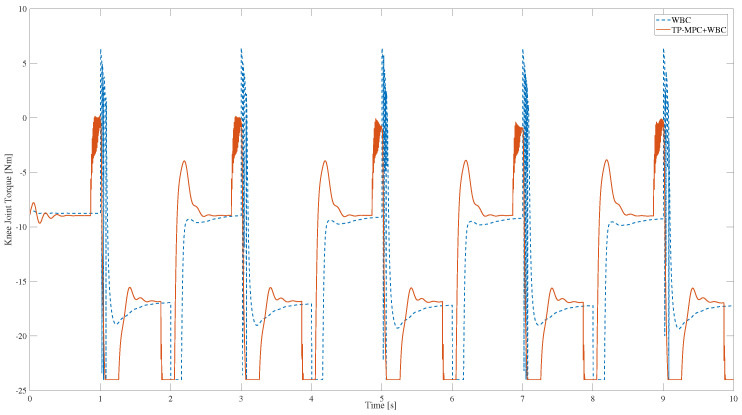
The knee torque solved by QP in WBC.

**Figure 7 sensors-25-00435-f007:**
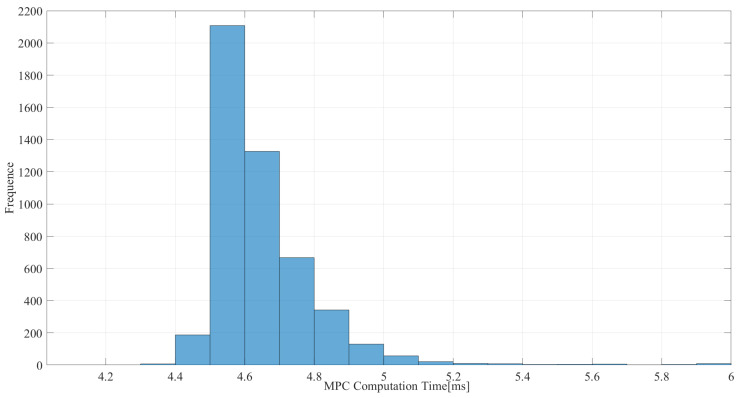
Time taken in TP-MPC with a square wave trajectory.

**Figure 8 sensors-25-00435-f008:**
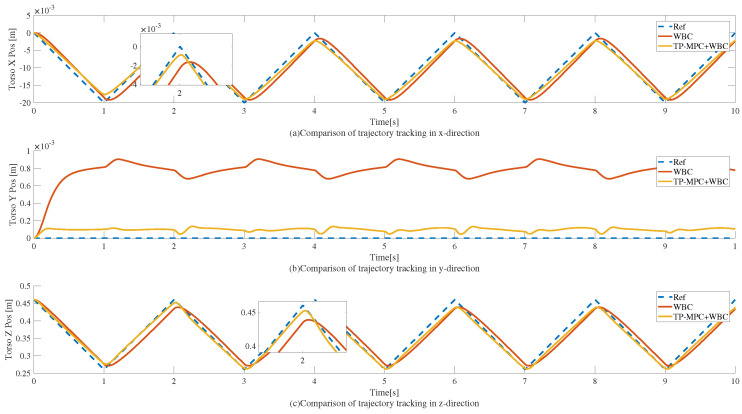
Trajectory tracking of a humanoid robot squatting motion when the reference trajectory is a triangle wave. Here, (**a**–**c**) represent the trajectories in the x, y, and z directions, respectively.

**Figure 9 sensors-25-00435-f009:**
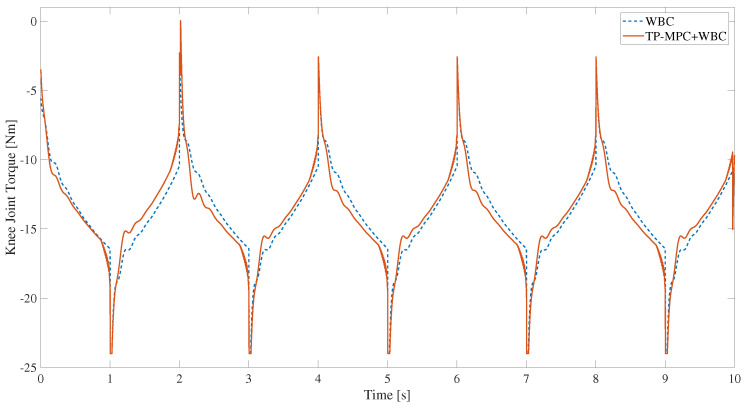
The knee torque solved by QP in WBC.

**Figure 10 sensors-25-00435-f010:**
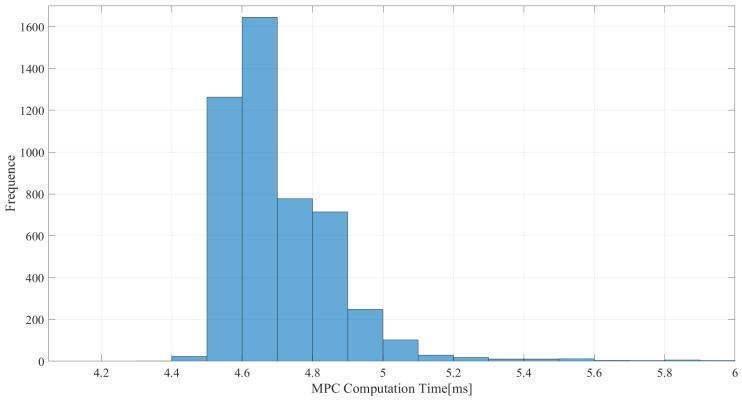
Time taken in TP-MPC with a triangle wave trajectory.

**Table 1 sensors-25-00435-t001:** Weight matrix for each term in TP-MPC.

Weight Term	Weight Matrix
$W_{CoM\;pos}$	diag($7.5\times 10^{2}$, $1.5\times 10^{6}$, $3.2\times 10^{2}$)
$W_{CoM\;vel}$	diag(7.0, $1.0\times 10^{1}$, 2.5)
$W_{left\;arm}$	diag($1.0\times 10^{3}$, $1.0\times 10^{1}$, $1.0\times 10^{2}$, $1.0\times 10^{3}$, $1.0\times 10^{1}$, $1.0\times 10^{2}$)
$W_{right\;arm}$	diag(1.0, $1.0\times 10^{1}$, 2.5, 1.0, $1.0\times 10^{1}$, 2.5)
$W_{u}$	diag($1.0\times 10^{-3}$, 1.0, $1.0\times 10^{-5}$, 1.0, 1.0, 1.0, 1.0, 1.0, 1.0)

**Table 2 sensors-25-00435-t002:** Tasks, Constraints for squat motion.

Tasks	Constraints
Torso trajectory tracking task	Floating-base dynamic
Arm trajectory tracking task	Friction cone
Feet trajectory tracking task	Joint torque
Wrench task	Joint velocity
Chest joint immobilization task	Joint power

**Table 3 sensors-25-00435-t003:** Weight matrix for each subtask in WBC.

Weight Term	Weight Matrix
$W_{torso}$	diag(1750, 450, 300, 70, 150, 50)
$W_{arm}$	diag( 300, 300, 300, 200, 200, 200)
$W_{feet}$	diag(900, 900, 900, 190, 140, 150)
$W_{wrench}$	diag(1, 1, 1, 2, 2, 2)
$W_{chest\;joint}$	diag(70)

**Table 4 sensors-25-00435-t004:** Tracking error at square wave.

Item	TP-MPC + WBC	WBC
Torso X Pos ($m^{2}$ · $10^{-5}$)	0.98	4.6
Torso Y Pos ($m^{2}$ · $10^{-7}$)	0.46	4.5
Torso Z Pos ($m^{2}$ · $10^{-3}$)	1.1	5.8

**Table 5 sensors-25-00435-t005:** Tracking error with a triangle wave trajectory.

Item	TP-MPC + WBC	WBC
Torso X Pos ($m^{2}$ · $10^{-6}$)	1.1	3.7
Torso Y Pos ($m^{2}$ · $10^{-7}$)	0.1	6.2
Torso Z Pos ($m^{2}$ · $10^{-4}$)	1.2	3.2

## Data Availability

No new data were created or analyzed in this study. Data sharing is not applicable to this article.

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
