# Peer review of "Squat Motion of a Humanoid Robot Using Three-Particle Model Predictive Control and Whole-Body Control"

_sensors, 2025, doi:10.3390/s25020435_

Round 1

Reviewer 1 Report

Comments and Suggestions for Authors

The paper presents a significant advancement in the field of bipedal balance control for humanoid robots, focusing specifically on the implementation of a squatting task. Through the use of dynamics software for simulations, the authors have developed a novel three-mass model that simplifies the representation of the humanoid robot during the squatting motion. This model facilitates the online optimization of the squatting trajectory using Model Predictive Control (MPC), which is then integrated into the Whole-Body Control (WBC) system of the robot. The results demonstrate effective control performance, highlighting the practical implications of the proposed methods.

While the technical contributions of the paper are noteworthy and possess considerable application value, the writing quality requires improvement. There are several grammatical issues and awkward phrasings that detract from the overall clarity of the work. The authors are encouraged to revise the manuscript to enhance its readability and coherence.

In conclusion, despite the need for better English writing, the innovative approaches and positive results warrant the paper's acceptance for publication in the journal Sensors.

Comments on the Quality of English Language

The paper presents a significant advancement in the field of bipedal balance control for humanoid robots, focusing specifically on the implementation of a squatting task. Through the use of dynamics software for simulations, the authors have developed a novel three-mass model that simplifies the representation of the humanoid robot during the squatting motion. This model facilitates the online optimization of the squatting trajectory using Model Predictive Control (MPC), which is then integrated into the Whole-Body Control (WBC) system of the robot. The results demonstrate effective control performance, highlighting the practical implications of the proposed methods.

While the technical contributions of the paper are noteworthy and possess considerable application value, the writing quality requires improvement. There are several grammatical issues and awkward phrasings that detract from the overall clarity of the work. The authors are encouraged to revise the manuscript to enhance its readability and coherence.

In conclusion, despite the need for better English writing, the innovative approaches and positive results warrant the paper's acceptance for publication in the journal Sensors.

Author Response

Thank you very much for reviewing our manuscript and for your valuable comments, we have rechecked the wording and re-corrected it according to your comments.

Reviewer 2 Report

Comments and Suggestions for Authors

The paper deals with the control of the squatting motion of humanoid robots. The authors modelled the humanoid robot as a three-mass model, and optimized the trajectory of the squatting motion based on this model.  Then, an MPC-WBC method is proposed to achieve the squatting motion of the robot. Finally, the proposed algorithm is verified by simulations results.  The research method in this paper is technically sound and the conclusions are credible. The reviewer suggests acceptance with minor revisions.

 The authors should consider the following comments:

1) The motivation of the research should be addressed more clearly.

2) The results of the simulations should be compared with relevant studies in the field.

3) Please be careful when using the word "first" to evaluate your own work.

4) Please provide the full name of the acronym if it appears for the first time.

Author Response

We are extremely grateful for the time and effort you have dedicated to reviewing our manuscript. Your valuable comments and suggestions are of great significance to us and have provided crucial guidance for improving our research work. We have carefully studied each comment and are now providing our detailed responses and the revised manuscript as follows.Please see the attachment.

Reviewer 3 Report

Comments and Suggestions for Authors

The TP-MPC combined with WBC is applied to a humanoid robot in the manuscript. The arms, legs, and torso are simplified into three separate COMs and TP-MPC is used to optimize the planned reference trajectory, while WBC is used to make the trajectory tracking of the optimized trajectory. The algorithm is tested by implementing a continuous squatting motion in simulation on a humanoid robot., The proposed TP-MPC+WBC assures more accurate trajectory tracking in comparison to to WBC only. The proposed algorithm optimizes the phenomenon of excessive spikes in the knee torque and is able to operate at a frequency of 100 Hz.

The topics of the paper is interesting for the specialists involved to control algorithms of humanoid robots.

It was very difficult to follow the derivations of equations in the text. I found some grammar and other mistakes in the manuscript and also give you some recommendations how to improve the text to be more readable:

1) line 6: The abbreviation WBC-WQP doesn't match with the text nearby.

2) line 12: The sentence: "...algorithm is less computationally time consuming...". By my opinion: you should say this if the execution time would be less than 2 ms and not 10 ms (100 Hz) as you mentioned in the abstract and later in the text of manuscript. Try to find information in other papers if for the squatting is 100 Hz enough?

3) Line 26 and 27: The abbreviation LIP and SLIP are not explained.

4) Line 28 and the lines 98 to 107: It is highly unusual to mention nationality and name of the institution in the scientific paper, it is enough to make only a simple citation.

5) Line 72: When the sentence is finished with the dot, than the next sentence start with capital letter.

6) Due to the strict rules of the journal MDPI Sensors  (see Instructions for Authors) how to organize the sections in the paper, I recommend to merge Section 2 and 3 and also section 4.1 into a section 2. Methods. Section 4.2 should be split into sections: 3. Results and 4. Discussion, while your section 5. Conclusions should be incorporated into new section 4.Discussion.

7) Line 137: Variable "l" should be "l hip"!

8) Line 144: Probably the grammar mistake: instead of "...are used as and control inputs ..." should be written: "..are used as control inputs ..."!

9) Line 146: there is no descriptions for all variables in inside the brackets!

10)  Lines 148 and 157: Are the elements E in the matrix of the eq. (12) the same as the element E in eq. (6)? Probably not!

11) Line 169: Please, describe more precisely the H and g terms  with equations, there is not enough good explanation in the text in lines 169 and 170.

12) Line 171, line 261-264: You have to describe in the nowadays subsection 4.1 what is the qp0ASS library, Webots software and RBDL library and move at least first paragraph of this section to the beginning of newly formed section 2. Method.

13) Line 173: The sentence must be finished with dot (.), not by comma and dot. (,.).

14) Line 137, Fig. 2: What is WQP in Fig. 2?

15) Line 204: Explain the term I 6x6 in the Equation (19).

16) Line 232: What is M in Equation (19)?

17) Lines 234 to 259: I am sorry, but du to poor explanation I was unable to understand all subsections 3.4.2 to 3.4.5. Try to rewrite all mentioned subsections to be readable. The main problem is the lack of explanations of some variables.

18) Line 307, caption of Fig. 6:  I don't understand what does mean: "... by QP in WBC."!

19) Lines 311 and 312: There should be no dot between words "figure" and   "(40)"!

20) Line 316: There is missing a space between "... value." and "From"!

21) Lines 326, 327, 342 and Figure 9: Use word "torque" instead of "moment".

22) Line 293, Figure 5: Would you explain why the yellow curve (TP-MPC) start to react (change) before the reference curve?

Comments on the Quality of English Language

The English should be improved! There is a lot of typewritten mistakes. Some of the I listed to the authors.

Author Response

We are extremely grateful for the time and effort you have dedicated to reviewing our manuscript. Your valuable comments and suggestions are of great significance to us and have provided crucial guidance for improving our research work. We have carefully studied each comment and are now providing our detailed responses and the revised manuscript as follows。Please see the attachment.

Reviewer 4 Report

Comments and Suggestions for Authors

The authors use three-particle model predictive control (TP-MPC) to optimise the reference trajectory and weight-based whole body control (WBC-WQP) to optimise the trajectory tracking capability for the humanoid robot to achieve continuous squatting action, and finally simulation experiments are carried out on the square and triangular wave reference trajectory in the Webots software, and the experimental results show that the algorithm TP-MPC in combination with WBC is able to improve the trajectory tracking capability, reduce the joint moments and improve the computational efficiency. The experimental results show that the algorithm TP-MPC combined with WBC can improve the trajectory tracking ability, reduce the joint moments and improve the computational efficiency. However, it is suggested that the following issues need to be revised before publication:

1. In the Introduction 1.4 Contribution, the author merely lists the work done in the text without reflecting the innovative nature of the work, so I think the author should add the appropriate details to clearly show the merits of the work.

2. In 2.1 Modelling, the author mentions that three particles refer to two arms, two legs and the torso, and calls this classification a three-particle model, so I think the author should improve the relevant details as to what the advantages of this model are compared to other models, and whether there is any relevant literature that can prove that this model has a stronger ability to optimise the trajectory of robots.

3. In Eq. (13) and Eq. (18), the authors set up the TP-MPC cost function and WBC optimisation model respectively, and there are a large number of unknown parameters in the model, which often determine the stability of the system and the performance of the simulation, but the authors didn't explain the sources of the parameters or give the specific set values. In order to ensure the feasibility of the simulation results, so I think the authors should improve the details of the parameters.

4. The author modelled the robot dynamics in 3.1 Modeling, but in the actual simulation the software's own dynamics program is used to deal with the problem, so I think the author should delete that part.

5. The authors have expressed the workflow of the PD controller in Eq. (20), but there are differences between this part and the PD controller in the control block diagram of Fig. 2, and the authors should make changes.

6. The author used the TP-MPC algorithm to generate the optimised trajectory in the first paragraph of 4.2 Results, but the trajectory curve in the Y-direction in Fig. 4 shows significant high frequency fluctuations, then I believe that the trajectory is not reasonable as an optimised result and should be checked by the author.

7. There are several formatting and formula editing problems in the text, such as the lack of a detailed description of the corresponding symbols in formula 4, the lack of centre alignment of the matrix elements in formula 27, and the writing of the bracket symbols in formula (1) as matrix symbols, etc., which the authors should revise.

Author Response

(The authors gave the same response as above.)

Round 2

Reviewer 3 Report

Comments and Suggestions for Authors

I am satisfied with answers, explanations and corrections done by authors of the manuscript. So, I have no more comments!

Author Response

Dear Reviewer, I am extremely honored to receive your positive feedback on my manuscript. It is truly a great encouragement and affirmation for me. During the writing process, I have been committed to rigorous research and clear exposition. Now, being recognized from your professional perspective makes me feel that all the previous efforts have paid off. Your meticulous review comments provided crucial guidance for the improvement of the manuscript, enabling me to optimize the paper from different dimensions and make it more scientific and readable. Thank you again for taking the precious time to review my work. Your professional spirit has always inspired me to move forward on the academic path. If there is an opportunity to submit to your journal in the future, I sincerely hope to continue to receive your guidance. Wish you all the best and fruitful academic achievements!   Best regards,

Reviewer 4 Report

Comments and Suggestions for Authors

Please use the standard expression format in Table 1.

I have no more suggestions for this paper.

Author Response

Thank you for pointing out the problem. Please see the attachment.
